# High prevalence of fecal carriage of Extended-spectrum beta-lactamase and carbapenemase-producing Enterobacteriaceae among food handlers at the University of Gondar, Northwest Ethiopia

**Azanaw Amare**[ID]*, **Setegn Eshetie**[ID], **Desie Kasew, Feleke Moges**

Department of Medical Microbiology, School of Biomedical and Laboratory Sciences, College of Medicine and Health Sciences, University of Gondar, Ethiopia

* azanaw03@gmail.com

## Abstract

### Background

Fecal carriage of extended-spectrum beta-lactamase and Carbapenemase-producing Enterobacteriaceae is a potential risk for the transmission of infection with resistant strains. Understanding the burden of these resistant strains in asymptomatic people is essential to reduce the chain of infection transmission. However, data on the fecal carriage of Extended-spectrum Beta-lactamase and Carbapenemase-producing Enterobacteriaceae among food handlers were limited in developing countries especially in Ethiopia. The aim of the present study is, therefore, to assess fecal carriage rate, associated factors, and antimicrobial resistance patterns of Extended-spectrum Beta-lactamase and Carbapenemase-producing Enterobacteriaceae among food handlers at the University of Gondar Cafeterias, Northwest Ethiopia.

### Materials and methods

An institution-based cross-sectional study was conducted from February to June 2021 at the University of Gondar cafeterias. A total of 290 stool samples were collected, transported using Cary Blair transport medium, and processed. All isolates were cultured and identified by using MacConkey agar, and routine biochemical tests. Antimicrobial susceptibility testing was done to each isolate following the Kirby Bauer disk diffusion method. If the zone of inhibition was ≤ 22 mm for ceftazidime, ≤25 mm for ceftriaxone, and ≤27 for cefotaxime they were considered as potential ESBL strain and selected for a further phenotypic confirmatory. Moreover, the double-disc diffusion test and the modified carbapenem inactivation method were used for confirmations of Extended-spectrum β-lactamase and Carbapenemase-producing Enterobacteriaceae respectively. If a ≥5mm difference in zone diameter for either antimicrobial agent in combination with clavulanic acid versus the zone diameter of the agent when tested alone (without B-lactamase inhibitor), was confirmed as ESBL-PE

**Data Availability Statement:** All relevant data are within the paper and its Supporting information files.

**Funding:** The author(s) received no specific funding for this work.

**Competing interests:** The authors have declared that no competing interests exist.

and if the zone of inhibition diameter between 6-15mm and 16- 18mm with a pinpoint colony, it was considered as carbapenem resistance Enterobacteriaceae. Data were entered using Epi-data version 4.6 and then exported to SPSS version 26 for analysis. Potential risk factors were assessed using multivariable logistic regression and a p-value less than 0.05 was considered statistically significant.

## Results

Out of 290 stool samples, 63 (21.7%) and 7 (2.4%) were confirmed as Extended-spectrum β-lactamase and Carbapenemase-producing Enterobacteriaceae. The most predominant ESBL-PE was *E. coli* 43 (14.8%) followed by *K. pneumoniae* 17 (5.9%). Most of the Extended-spectrum β-lactamase and Carbapenemase-producing isolates were resistant to tetracycline, cefotaxime, ceftazidime, and ceftriaxone (100% each). In contrast, a low resistance level was recorded for Meropenem and cefoxitin. The overall Multi-drug resistant Enterobacteriaceae (MDR) was 147 (42.3%). Antibiotics usage in the last 3 months and drinking unpasteurized milk were associated with the carriage of the Extended-spectrum beta-lactamase-Producing Enterobacteriaceae.

## Conclusions and recommendations

The high fecal carriage rate of Multi-drug resistance isolate, Extended-spectrum β-lactamase, and Carbapenemase-producing Enterobacteriaceae were recorded among food handlers. Therefore, this study gives signals in the spread of drug-resistant bacteria easily to the community. Hence, the need for adjusting and promotion of infection prevention measures to prevent the spread of drug-resistant bacteria should not be underestimated.

## Introduction

Enterobacteriaceae are Gram-negative, facultative anaerobes, and non-spore-forming bacilli. It is mainly found in the gastrointestinal tract of humans and is a significant cause of nosocomial and community-acquired infections. Mostly they cause urinary tract infection, respiratory tract bloodstream, and wound infections. Most *Enterobacteriaceae* are treated by different antimicrobial agents, but if it is improperly used which leads to multidrug-resistant (MDR) or extensively drug resistance (XDR), The rising trend of antimicrobial resistance has become a global problem which is mainly caused by Enterobacteriaceae [1, 2]. Those Enterobacteriaceae are the most important emerging resistance characteristics which correspond to resistance for β-lactam and carbapenems antibiotics [3, 4].

 *Escherichia coli* (*E. coli*) and *Klebsiella pneumoniae* (*K. pneumoniae*) are two prevalent Enterobacteriaceae species with pathogenic potential and ESBL-encoding genes [5]. The Infectious Diseases Society of America has listed them as two out of six pathogens for which new drugs are urgently needed to tackle resistance development [6]. Infections with ESBL-producing bacteria are linked to nearly double the mortality rate as infections with non-ESBL producers, according to a meta-analysis [7]. Colonization with ESBL-producing bacteria is a cause for concern even when there is no infection present. Carriage of resistant commensal Enterobacteriaceae strains in the normal gut flora may act as a reservoir for resistance genes that are then acquired by pathogenic strains [8]. *E. coli* and *K. pneumoniae* virulence determinants are

adhesins, toxins, iron acquisition factors, lipopolysaccharides, polysaccharide capsules, types 1 and 3 fimbriae, and invasins are typically encoded on pathogenicity islands (PAIs), plasmids, and other mobile genetic components [9, 10].

Multidrug resistance has been increased all over the world that is considered a public health threat. Several recent investigations reported the emergence of MDR bacterial pathogens from different origins including humans, birds, cattle, and fish that increase the need for routine application of the antimicrobial susceptibility testing to detect the antibiotic of choice as well as the screening of the emerging MDR strain [11–18].

Extended-Spectrum Beta-Lactamase (ESBL) and Carbapenemase are enzymes usually produced by micro-organisms in the gut, such as *E. coli* and *Klebsiella spp*. The enzymes produced by resistant bacteria can destroy certain antibiotics and render them useless, making the treatment of infections more difficult. Such bacteria can easily be transmitted from person to person, such as by fecal contamination or by touching contaminated equipment or the air of the ward [19, 20]. ESBL-producing Enterobacteriaceae (ESBL-PE) and Carbapenemase-producing Enterobacteriaceae (CPE) are increasing both in hospital and community settings [21, 22].

As drug resistance widely spreads, a major concern is the coexistence of multiple ESBL and Carbapenemase genes that have led to the emergence of organisms that are resistant to nearly all antibiotics [23]. Inappropriate and irrational use of antimicrobial drugs, poor sanitation, and infection control practices in developing countries play a critical role in an increased prevalence of resistant bacteria in a community providing favorable conditions for resistant microorganisms to emerge and spread [24, 25]. This can lead to a proliferation of organisms with the broad-spectrum β-lactamase activity that threatens the future of the βlactam class in clinical care [24].

The reports showed that an increase in the carriage of antimicrobial-resistant bacteria in the community increases the spread of resistant bacteria via human-to-human transmission. In addition, resistance bacteria can be spread from one person to another in healthcare settings through contaminated hands, surfaces, and also through contaminated food or water [26, 27]. Infections related to extended-spectrum beta-lactamase and Carbapenemase-producing Enterobacteriaceae represent a major global health threat [28].

However, such data are limited in food handlers in Ethiopia; therefore, we designed this study aimed to assess fecal carriage rate, associated factors, and antimicrobial resistance patterns of Extended-spectrum Beta-lactamase and Carbapenemase-producing Enterobacteriaceae among food handlers at the University of Gondar Cafeteria.

## Materials and methods

### Study design, period, and area

An institution-based cross-sectional study was conducted from February to June 2021 at the University of Gondar cafeterias. The University of Gondar is located in Gondar town, 737 km from Addis Ababa and 180 km from Bahir Dar in Northwest Ethiopia. The Department of Medical Microbiology has four Laboratory sections which include Bacteriology, Mycology, Virology, and Tuberculosis and Leprosy Laboratory. In the Bacteriology section culture is one of the main activities including bacterial isolation, identification, and antimicrobial susceptibility test. At present, the University of Gondar has approximately 27,000 students in regular and extension programs. In the University there are six cafeterias (including Hospital, College of Medicine and Health Sciences, Atse Tewodros, Maraki, Atse Fasile, and Tseda). Currently, those cafeterias serve meals for 22,000 students and more than 600 patients. According to the information obtained from human resource management of the University of Gondar, approximately 650 food handlers are serving in these cafeterias.

**Study population, sample size, and sampling technique.** The source population was all food handlers who were working at the University of Gondar cafeteria. The study populations were asymptomatic food handlers who were working at the University of Gondar cafeteria food establishment and available during the data collection period and voluntary to participate in the study. Food handlers who had been on antibiotic treatment during data collection were excluded from this study. Food handlers who had been working in the six campuses cafeteria of the University were included in the study. The sample size was calculated based on a single population proportion formula and the calculated sample size was 290. The value of proportion was taken as 25.3% (0.253) from the previous study conducted by Diriba *et al.*, 2020, on "Fecal carriage rate of extended-spectrum beta-lactamase-producing *Escherichia coli* and *Klebsiella pneumoniae* among apparently health food handlers in Dilla University student cafeteria" Southwest Ethiopia [29]. A stratified random sampling technique was used to recruit 290 study participants from the sampling frame of food handlers. To select representative participants, the final sample size was proportionally allocated to each stratum.

**Data collection and analysis.** Data related to socio-demographic, clinical, and hygiene-related data were collected by face-to-face interviews of the food handlers using a well-structured questionnaire before laboratory sample collection. All data were collected and analyzed by two trained laboratory technologists.

**Sample collection and processing.** The study participant**s** were instructed to collect approximately 2 grams of stool into a clean, leak-proof container. Following collection, all stool specimens were labeled with sample number, time, and date of sample collection. Each stool sample was transported to the School of Biomedical and Laboratory Sciences, Medical Microbiology laboratory section using Cary-Blair transport media (HiMEDIA Laboratories Pvt. Ltd., Mumbai, India). A loop full of stool sample was inoculated on MacConkey agar (Oxoid Ltd, Basingstoke, UK) using the Streak Plate Method using an aseptic technique and incubated aerobically at 37˚C for 18–24 hours [30].

**Bacterial isolation and identification.** *Cultural observation*. Preliminary identification of bacteria was done based on colony characteristics of the organisms. Some colony characteristics like (size, shape, color, pigmentation, texture, elevation, and edge [30].

*Microbiological analysis of stool specimen*. Enterobacteriaceae were identified by inoculating /streaking of stool samples on MacConkey agar (Oxoid Ltd, Basingstoke, UK) based on their color morphology after an incubation time of 18–24 hours at 37˚C. The smear was prepared from each different colony observed on the plates and Gram staining was performed. The Results such as gram reaction (gram-negative), arrangements, and shape of bacteria are seen from the examinations using a microscope [31].

*Biochemical examination*. Biochemical tests were performed on colonies from pure cultures for the identification of the isolates. Triple sugar iron agar (Oxoid Ltd, Basingstoke (for gas production, lactose fermentation, and hydrogen sulfide production), UK), indole test (for tryptophan utilization, Simon's citrate agar (Oxoid Ltd, Basingstoke, UK) (citrate utilization test), urease agar (Oxoid Ltd, Basingstoke, UK) (urease production test), lysine iron agar (Oxoid Ltd, Basingstoke, UK) (lysine decarboxylase test), and Motility medium (Oxoid Ltd, Basingstoke, UK) (motility test) were included in the biochemical tests for species identification [32].

**Antimicrobial susceptibility testing.** Following bacterial identification, the antimicrobial susceptibility testing (AST) of the isolates was performed by a modified Kirby-Bauer disk diffusion technique by following the Clinical and Laboratory Standard Institute (CLSI) guideline, 2020 [33]. To make bacterial suspension the pure colonies of a young culture were picked using a sterile wire loop and emulsified in 0.85% sterile normal saline and compared with 0.5 McFarland turbidity standard. Then the bacterial suspension was inoculated onto Muller-Hinton agar (Oxoid, Basingstoke, and Hampshire, UK) using the lawn culture method.

The following antibiotics disks were used as β-lactam combination group (Amoxicillin/cla-vulanic acid (AUG 20/10 μg), Beta-lactams (Cefotaxime (CTX 30 μg), Ceftazidime (CAZ 30 μg), Ceftriaxone (CRO 30 μg), and Cefoxitin (CXT 30 μg)), Aminoglycosides (Gentamicin (GEN 10 μg)), Carbapenems (Meropenem (MER 10 μg)), Tetracycline (Tetracycline (TE 30 μg), Fluoroquinolones (Ciprofloxacin (CIP 30 μg)), Phenicols (Chloramphenicol (CHL 30μg)), and Sulfonamide (Trimethoprim/Sulphamethoxazole (23.75 μg /1.25μg)). The antibi-otic disks used were from BD, BBL$^{TM}$ Company, USA Product. Then the plates were incubated at 37 °C for 18–24 hours. After overnight incubation, the zone of inhibition was measured and interpreted as susceptible, intermediate, and resistant based on the recommendation of CLSI, 2020 [33]. Multi-drug resistance patterns of the isolates were identified using the criteria set by Magiorakos et.al. [34].

**Detection and confirmation of extended-spectrum β- lactamase (ESBL).** All strains were tested against ceftriaxone, cefotaxime, and ceftazidime for ESBL screening using the Kirby-Bauer disk diffusion method. If the zone of inhibition was ≤ 22 mm for ceftazidime, ≤25 mm for ceftriaxone, and ≤27 for cefotaxime they were considered as potential ESBL strain and selected for a further phenotypic confirmatory test as described below [33]. Pheno-typic confirmation of ESBL production was done using the double-disc diffusion test and interpreted by following the CLSI, 2020 guidelines. Pure culture colonies of ESBL-PE isolates were emulsified in 0.85% saline and compared with 0.5 McFarland turbidity standard then inoculated by lawn culture method using sterile swabs on MHA. The following antibiotic disks such as cefotaxime (30μg), cefotaxime/clavulanic acid (30μg/10μg), ceftazidime (30μg), and ceftazidime/clavulanic acid (30μg/10μg), were used as recommended by CLSI, 2020 to estab-lish the status of the ESBL phenotypes. The plates were then incubated aerobically at 37˚C for 18 to 24 hours. *K. pneumonia*e American Type Culture Collection (ATCC) 700603 (positive control) and *E. coli* strain ATCC 25922 (negative control) were used for quality control. If a ≥5mm difference in zone diameter for either antimicrobial agent in combination with clavulan-ic acid versus the zone diameter of the agent when tested alone (without B-lactamase inhibi-tor), was confirmed as ESBL-PE [33, 35].

**Detection and confirmation of Carbapenemase-producer Enterobacteriaceae.** Carba-penemase-producing Enterobacteriaceae was screened by using Meropenem disks. The sus-pension of isolated bacteria was inoculated onto MHA, then Meropenem (10μg) disks were placed and incubated at 37˚C for18- 24 hrs. If the zone of inhibition is ≤ 19 mm, it was consid-ered as a presumptive CPE [33]. The suspected CPE was confirmed by the Modified carbape-nems inactivation methods (mCIM) and EDTA-modified carbapenem inactivation method (eCIM). The isolates of a bacterial colony that was suspected for CPE were diluted with 2 ml of trypticase soya broth and the Meropenem (10μg) disk was immersed in the suspension and 0.5 M EDTA (only for eCIM) was added; then incubated for 4 hours. A pure colony standard strain of Meropenem susceptible *E. coli* ATCC 25922 was emulsified in 0.85% normal saline and compared with 0.5 McFarland standard then inoculated the whole plate of MHA. After 4 hours of incubation, the Meropenem disk was removed from the test tube and placed on the MHA plate which was inoculated by *E. coli* ATCC 25922 Meropenem sensitive strain and incubated at 37˚C for 18–24 hours. After incubation, if the zone of inhibition diameter between 6-15mm and 16- 18mm with a pinpoint colony, it was considered as carbapenem resistance Enterobacteriaceae [33].

**Quality control.** The sterility of newly prepared culture media was checked by incubating 5% of the batch at 35–37˚C overnight before it was used and was evaluated for possible growth or contamination. The performance testing was done by inoculating known control strains of *E. coli* ATCC 25922 to check the quality of the culture media and antibiotics disks. For the ESBL confirmatory test, *K. pneumoniae* ATCC® 700603 (ESBLs positive) and *E. coli* ATCC®

25922 (ESBLs negative) control strains were used. *K. pneumoniae* ATCC 1705 and ATCC 1706 were used as positive and negative quality control for Carbapenemase production respectively [33].

**Data processing and analysis.** All data were checked for completeness, coded, and entered using Epi-data version 4.6 and exported to SPSS version 26 for statistical analysis. Then data were analyzed using SPSS version 26 to determine the independent variables and the fecal carriage rate of ESBL and CPE frequency analysis and cross tab were used. The Chi-square test was used with appropriate correction for the observation. To determine the associated factors Multivariable Logistic regression analyses were used. A variable with a p-value ≤ 0.2 in bivariate logistic regression was checked in multivariate analysis for a statistically significant association by controlling the possible confounders. Crude and adjusted odds ratios were used to quantify the strength of association between ESBL-PE and CPE carriage rate and risk factors. A variable with a p-value of less than 0.05 at a 95% confidence interval was considered statistically significant.

**Ethical approval.** Ethical clearance and support letter were obtained from the Ethical Review Committee of the School of Biomedical and Laboratory Sciences, College of Medicine and Health Sciences, the University of Gondar with reference number SBLS/02/25/2021. Each study participant was informed about the purpose, methods of collection, anticipated benefit, and risk of the study. Written informed consent was obtained from each study participant. Food handlers who were found to be ESBL and CPE positive for Enterobacteriaceae were referred to their respective staff medical center for appropriate antimicrobial treatments and health education was given about infection prevention.

## Results

A total of 290 food handlers were involved in this study. Over three fourth study participants were females 242 (83.4%). The majority of the study participants were between the age group of 20 and 40 years (82.8%) with a mean age of 32.10 (Standard deviation = ±8.691 years). Regarding educational status 98 (33.8%) completed primary school. Respondents with a family size of 1–5 were 262 (90.3%) and of the total respondents, 116 (40%) did not know the principle of food safety (Table 1).

Of the total respondents, 181 (62.4%) had periodic medical checkups, 93 (32.1%) of them used antibiotics without prescription, 97 (33.4%) had a history of diarrhea in the last three months, 58 (20%) had a history of urinary tract infection in the last three month, and 129 (45.5%) of them had antibiotics used in the last 3 months. About 71 (24.5%) of the study participant drink unpasteurized milk, while 59 (20.3%) and 148 (51%) of them had the habit of eating raw meat and raw vegetables respectively (Table 1).

### Prevalence of Enterobacteriaceae isolates from the stool sample

All of the study participants had culture-positive results and a total of 347 bacterial isolates were identified. The study revealed that 233 of the study participants contained single organisms 57 of the study participants contained two organisms. Out of 347 bacterial isolates, *E. coli* were the most prevalent 245 (70.6%) followed by *K. Pneumoniae* 68 (19.6%) and *C. freundii* 13 (3.7%) (Fig 1).

**Phenotypic characteristics of the recovered isolates.** *Escherichia Coli is* a bacteria that are gram-negative, rod-shaped, and motile, grows on MacConkey agar (red or colorless colonies), lactose fermenter, indole positive, urease negative, Citrate negative, gas producer, hydrogen sulfide negative, and lysine decarboxylase positive.

**Table 1. Socio-demographic and clinical characteristics, and hygienic practices of food handlers at the University of Gondar Comprehensive Specialized Hospital and Student Cafeteria, Northwest Ethiopia, February to June 2021.**

| Variables | Category | Frequency | Percentage (%) |
|---|---|---|---|
| Gender | Male | 48 | 16.6 |
| | Female | 242 | 83.4 |
| Age in years | <20 | 8 | 2.8 |
| | 20–40 | 240 | 82.8 |
| | >40 | 42 | 14.4 |
| Level of education | Illiterate | 57 | 19.7 |
| | Primary | 98 | 33.8 |
| | Secondary and above | 135 | 39.7 |
| Monthly income | Less than 1000 ETB | 7 | 2.4 |
| | Greater than 1500 ETB | 283 | 97.6 |
| Marital status | Married | 159 | 54.8 |
| | Unmarried | 131 | 45.2 |
| Family size | 1–5 | 262 | 90.3 |
| | >5 | 28 | 9.7 |
| Service years | 1–5 | 152 | 52.4 |
| | More than 5 | 138 | 47.6 |
| Knowledge of food safety practice | Yes | 174 | 60 |
| | No | 116 | 40 |
| Medical checkups | Yes | 181 | 62.4 |
| | No | 109 | 37.6 |
| History of medical instrumentation | Yes | 8 | 2.8 |
| | No | 282 | 97.2 |
| History of hospital admission in the last three months | Yes | 6 | 2.1 |
| | No | 284 | 97.9 |
| Used antibiotics without prescription | Yes | 93 | 32.1 |
| | No | 197 | 69.9 |
| History of diarrhea in the last 3 months | Yes | 97 | 33.4 |
| | No | 193 | 66.6 |
| History of urinary tract infection in the last 3 months | Yes | 58 | 20 |
| | No | 232 | 80 |
| Antibiotics used in the last 3 months | Yes | 129 | 45.5 |
| | No | 161 | 55.5 |
| History of chronic disease | Yes | 38 | 13.1 |
| | No | 252 | 86.9 |
| Hand washing habit | Yes | 264 | 91 |
| | No | 26 | 9 |
| Hands wash with soap | Only with water | 63 | 21.7 |
| | With soap | 227 | 78.3 |
| Fingernail status | trimmed | 273 | 94.1 |
| | not trimmed | 17 | 17 |
| Wear hair garment | Yes | 259 | 89.3 |
| | No | 31 | 10.7 |
| Source of water for drinking | pipe | 290 | 100 |
| | hand dug well | 0 | 0 |
| Pit latrine toilet facilities | Yes | 284 | 97.9 |
| | No | 6 | 2.1 |

*(Continued)*

**Table 1.** (Continued)

| Variables | Category | Frequency | Percentage (%) |
|---|---|---|---|
| Drinking unpasteurized milk | Yes | 71 | 24.5 |
| | No | 219 | 75.5 |
| Eating raw meat | Yes | 59 | 20.3 |
| | No | 231 | 79.7 |
| Eating raw vegetables | Yes | 148 | 51 |
| | No | 142 | 49 |

*Klebsiella pneumoniae* is a gram-negative, rod-shaped, grow on MacConkey agar (red large mucoid colonies) lactose-fermenting, indole negative, urease positive, Citrate positive, gas producer, hydrogen sulfide negative, non-motile, and lysine decarboxylase positive.

*Proteus mirabilis* is a gram-negative, rod-shaped bacteria that grow on MacConkey agar (colorless colony) non-lactose fermenter, indole negative, urease positive, motility positive, citrate positive, hydrogen sulfide positive, lysine decarboxylase positive.

*Citrobacter freundii* is a gram-negative rod-shaped that grow on MacConkey agar (Smooth, convex, translucent, or opaque grey colored with a shiny surface and entire margin; mucoid or rough colonies occasionally), lactose fermenter, indole negative, urease positive, H2S positive, motility positive, gas producer, lysine decarboxylase negative.

*Enterobacter cloacae* are Gram-negative, rod-shaped, grow on MacConkey agar, lactose fermenter, indole negative, urease negative, H2S negative, gas producer, citrate positive, motile, and lysine decarboxylase negative.

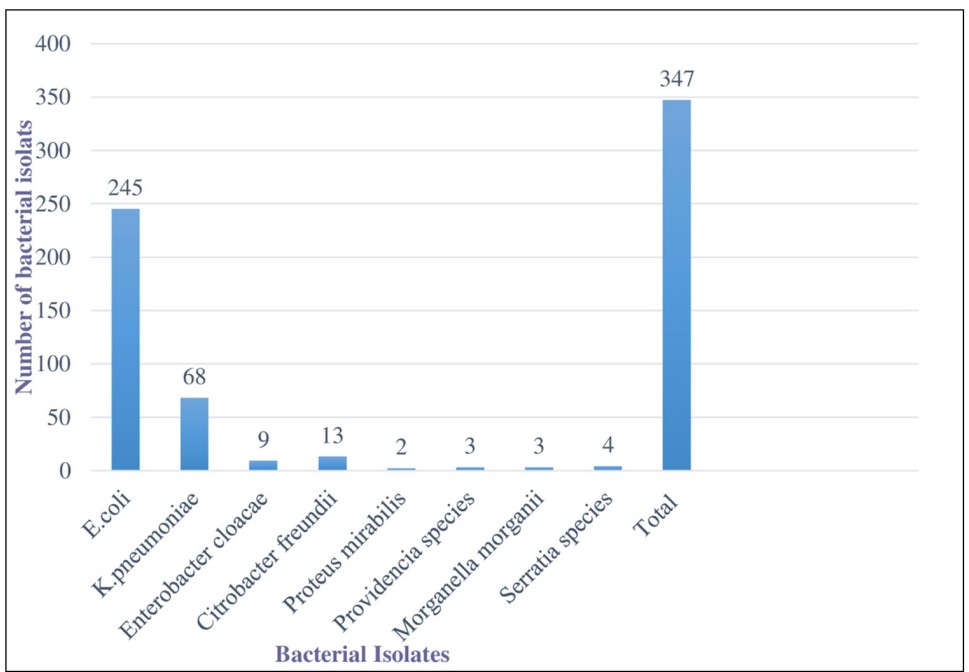

**Fig 1. Prevalence of Enterobacteriaceae isolates from stool sample at University of Gondar Comprehensive Specialized Hospital and students' cafeteria, February to June 2021.**

**Table 2. Antimicrobial resistance patterns of Enterobacteriaceae among food handlers at the University of Gondar Comprehensive Specialized Hospital and students' Cafeterias, Northwest Ethiopia, February to June 2021.**

| Classes | Antibiotics | E. coli (N = 245) | | K. pneumoniae (N = 68) | | E. cloacae (N = 9) | | C.freundii (N = 13) | | P.mirabilis (N = 2) | | Providencia species (N = 3) | | M.morganii (N = 3) | | Serratia species (N = 4) | | Total R (%) |
|---|---|---|---|---|---|---|---|---|---|---|---|---|---|---|---|---|---|---|
| | | S (N (%)) | R (N (%) | S (N (%) | R (N (%) | S (N (%) | R (N (%) | S (N (%) | R (N (%) | S (N (%) | R (N (%) | S (N (%) | R (N (%) | S (N (%) | R (N (%) | S N (%) | R (N (%) | |
| Aminoglycosides | GEN | 161 (65.7) | 84 (34.3) | 37 (54) | 31 (46) | 8(89) | 1(11) | 9 (69.2) | 4 (30.8) | 2(100) | 0 | 2(67) | 1(33) | 3(100) | 0 | 2(50) | 2(50) | 123 (35.5) |
| Beta-lactam combination group | AUG | 106 (43) | 139 (57) | 30 (44) | 38 (56) | 2(22) | 7(78) | 4(31) | 9(69) | 1(50) | 1(50) | 2(67) | 1(33) | 2(67) | 1(33) | 1(25) | 3(75) | 199 (57.3) |
| Beta-lactams | CAZ | 188 (77) | 57(23) | 51 (75) | 17 (25) | 9(100) | 0 | 9(69) | 4(31) | 2(100) | 0 | 2(67) | 1(33) | 3(100) | 0 | 2(50) | 2(50) | 81(23.3) |
| | CRO | 192 (78) | 53(22) | 50 (73) | 18 (27) | 7(78) | 2(22) | 10(77) | 3(23) | 1(50) | 1(50) | 2(67) | 1(33) | 2(67) | 1(33) | 3(75) | 1(25) | 80(23) |
| | CXT | 121 (90) | 24(10) | 65 (96) | 3(4) | 9(100) | 0 | 13 (100) | 0 | 2(100) | 0 | 3(100) | 0 | 3(100) | 0 | 4 (100) | 0 | 27(7.8) |
| | CTX | 196 (80) | 49(20) | 51 (75) | 17 (25) | 9(100) | 0 | 10(77) | 3(23) | 2(100) | 0 | 2(67) | 1(33) | 3(100) | 0 | 3(75) | 1(25) | 71(20.5) |
| Carbapenems | MER | 233 (95) | 12(5) | 62 (91) | 6(9) | 9(100) | 0 | 13 (100) | 0 | 2(100) | 0 | 3(100) | 0 | 3(100) | 0 | 4 (100) | 0 | 18(5) |
| Fluoroquinolones | CIP | 187 (76.3) | 58 (23.7) | 42 (62) | 26 (38) | 6(67) | 3(33) | 6 (46.2) | 7(53.8) | 2(100) | 0 | 3(100) | 0 | 2(67) | 1(33) | 2(50) | 2(50) | 97(28) |
| Phenicols | CHL | 208 (84.9) | 37 (15.1) | 62 (91) | 6(9) | 9(100) | 0 | 13 (100) | 0 | 2(100) | 0 | 3(100) | 0 | 3(100) | 0 | 3(75) | 1(25) | 44(12.7) |
| Sulfonamides | SXT | 117 (48) | 128 (52) | 28 (41) | 40 (59) | 8(89) | 1(11) | 4(31) | 9(69) | 1(50) | 1(50) | 1(33) | 2(67) | 2(67) | 1(33) | 1(25) | 3(75) | 185 (53.3) |
| Tetracycline | TE | 89(36) | 156 (64) | 28 (41) | 40 (59) | 8(89) | 1(11) | 3(23) | 10(77) | 0 | 2(100) | 2(67) | 1(33) | 2(67) | 1(33) | 2(50) | 2(50) | 212 (61.1) |

Key: S = Sensitive, R = Resistance, AUG = Augmentin/Amoxicillin/clavulanic acid; SXT = Sulphamethoxazole-trimethoprim; TE = Tetracycline; CIP = Ciprofloxacin; CHL = Chloramphenicol; GEN = Gentamycin; CXT = Cefoxitin; CRO = Ceftriaxone; CAZ = Ceftazidime; CTX = cefotaxime, MER = Meropenem.

## Antimicrobial resistance patterns of Enterobacteriaceae isolates

A total of eleven antibiotics from eight classes (beta-lactam, β-lactam combination group, Aminoglycosides, Carbapenems, Tetracycline, Fluoroquinolones, Phenicols, and Sulfonamides) of antimicrobials were used to assess the susceptibility patterns of the isolate. High resistance rates were observed for tetracycline (61.1%), Sulphamethoxazole-trimethoprim (53.3%), and amoxicillin/clavulanic acid (57.3%). In contrast, Meropenem and cefoxitin were the most effective antibiotics with sensitivity rates of (95%) and (92.2%) respectively (Table 2).

**Multi-drug resistant patterns of Enterobacteriaceae among food handlers.** The MDR determination was carried out by considering the classification of antibiotics in the following eight antibiotic classes (beta-lactam, β-lactam combination group, Aminoglycosides, Carbapenems, Tetracycline, Fluoroquinolones, Phenicols, and Sulfonamides). The overall MDR prevalence in this study was 147 (42.3%; 95% CI: 39.7–51). Accordingly, from the isolates of *C. freundii* were 9 (69.2%) MDR followed by *P. mirabilis* 1 (50%), *E. coli* 104 (42.4%), and *K. pneumoniae* 28 (41.2%) (Table 3). The MDR isolates across the six cafeterias were 42, 32, 35, 26, 6, and 6 from the Hospital cafeteria, College of Medicine and Health Science campus cafeteria, Maraki campus cafeteria, Atse Tewodros campus cafeteria, Atse Fasile campus cafeteria, and Tseda campus cafeteria respectively (Table 4).

**The fecal carriage rate of ESBL and Carbapenemase-producing Enterobacteriaceae.** From the total of 290 study participants, 63 (21.73%) (95% CI: 17.2–26.6) were colonized by ESBL-PE and 7 (2.4%) (95% CI: 0.7–4.5) with CPE. From the total bacterial isolates, 63

**Table 3. Multidrug resistance profiles of Enterobacteriaceae isolates among food handlers at the University of Gondar Comprehensive Specialized Hospital and Students 'Cafeterias, Northwest Ethiopia, February to June 2021.**

| Isolates N = 347 | Level of resistance n, (%) | | | | | | | | | | |
|---|---|---|---|---|---|---|---|---|---|---|---|
| | R0 | R1 | R2 | R3 | R4 | R5 | R6 | R7 | R8 | R9 | MDR$\geq$R3 |
| *E. coli* N = 245 | 42 | 46 | 53 | 36 | 20 | 10 | 14 | 16 | 5 | 3 | 104 (42.4) |
| *K.pneumoniae* N = 68 | 16 | 14 | 10 | 9 | 4 | 4 | 5 | 4 | 1 | 1 | 28 (41.2) |
| *E. cloacae* N = 9 | 1 | 1 | 6 | 1 | 0 | 0 | 0 | 0 | 0 | 0 | 1 (11) |
| *C. freundii* N = 13 | 2 | 1 | 1 | 3 | 0 | 1 | 3 | 1 | 1 | 0 | 9 (69.2) |
| *P. mirabilis* N = 2 | - | - | 1 | - | 1 | 0 | 0 | 0 | 0 | 0 | 1 (50) |
| Others N = 10 | 4 | 1 | 1 | 2 | 2 | 0 | 0 | 0 | 0 | 0 | 4 (40) |
| Total (N = 347) | 65 | 63 | 72 | 51 | 27 | 15 | 22 | 21 | 7 | 4 | 147 (42.7) |

Note; R0: sensitive for all classes of antibiotics, R1: resistant for one class of antibiotics, R2: resistant for two classes of antibiotics, R3: resistant for three classes of antibiotics, etc., MDR = multidrug resistant.

(18.2%) and 7 (2.01%) were confirmed to be positive for ESBL and Carbapenemase production respectively. Of these 63 ESBL isolates, 21, 16, 12, 7, 5, and 2 were found in the Hospital cafeteria, College of Medicine and Health Science campus cafeteria, Maraki campus cafeteria, Atse Tewodros campus cafeteria, Atse Fasile campus cafeteria, and Tseda campus cafeteria respectively. All CPE were 7 (100%) ESBL producers. Similarly, all ESBL producing isolates were MDR-E (Table 5).

**Table 4. Distribution of MDR-E isolates from the six cafeterias, among food handlers at the University of Gondar Comprehensive Specialized Hospital and Students' Cafeteria, Northwest Ethiopia, February to June 2021.**

| Study site | | Bacterial Isolates | | | | | | |
|---|---|---|---|---|---|---|---|---|
| | MDR Level | *E. coli* (n = 245) | *K. pneumoniae* (n = 68) | *C. freundii* (n = 13) | *E. cloacae* (n = 9) | *P. mirabilis* (n = 2) | Others *(n = 10) | Total N (%) |
| HC (N = 75) | Non-MDR N (%) | 23 (9.4) | 7 (10.3) | 1 (7.7) | 1 (11.1) | 1 (50) | - | 33 (44) |
| | MDR N (%) | 26 (10.6) | 10 (14.7) | 3 (23.1) | 1 (11.1) | 1 (50) | 1 (10) | 42 (56) |
| CMHSC (N = 55) | Non-MDR N (%) | 19 (7.8) | 3 (4.4) | 0 | - | - | 1 (10) | 23 (41.8) |
| | MDR N (%) | 21 (8.6) | 8 (11.8) | 2 (0.52) | - | - | 1 (10) | 32 (58.2) |
| MCC (N = 105) | Non-MDR N (%) | 49 (20) | 20 (29.4) | 0 | - | - | 1 (10) | 70 (66.7) |
| | MDR N (%) | 28 (11.4) | 5 (7.4) | 1 (7.7) | - | - | 1 (10) | 35 (33.3) |
| ATCC (N = 68) | Non-MDR N (%) | 28 (11.4) | 7 (10.3) | 2 (15.4) | 4 (44.4) | - | 1 (10) | 42 (61.8) |
| | MDR N (%) | 20 (8.2) | 2 (3) | 3 (23.1) | 0 | - | 1 (10) | 26 (39.2) |
| AFCC (N = 24) | Non-MDR N (%) | 10 (4.1) | 2 (3) | 1 (7.7) | 2 (22.2) | - | 3 (30) | 18 (75) |
| | MDR N (%) | 4 (1.63) | 2 (3) | 0 | 0 | - | - | 6 (25) |
| TsCC (N = 20) | Non-MDR N (%) | 12 (4.9) | 1 (1.5) | - | 1 (11.1) | - | - | 14 (70) |
| | MDR N (%) | 5 (2.1) | 1 (1.5) | - | 0 | - | - | 6 (30) |
| Total (N = 347) | Non-MDR N (%) | 141 (57.6) | 40 (58.8) | 4 (30.8) | 8 (88.9) | 1 (50) | 6 (60) | 200 (57.6) |
| | MDR N (%) | 104 (42.4) | 28 (41.2) | 9 (69.2) | 1 (11.1) | 1 (50) | 4 (40) | 147 (42.3) |

Key; HC = Hospital Cafeteria, CHMSC = College of Medicine and Health Sciences Campus Cafeteria, MCC = Maraki campus cafeteria, ATCC = Atse Tewodros Campus Cafeteria, AFCC = Atse Fasile campus cafeteria, and TsCC = Tseda campus cafeteria. Others* *M. morganii*, *Providencia* species, and *Serratia* species.

**Table 5. Distribution of ESBL-PE, Carbapenemase-producing Enterobacteriaceae, and MDR isolate among food handlers at the University of Gondar Comprehensive Specialized Hospital and Students' Cafeterias, Northwest Ethiopia, February to June 2021.**

| Study site | Isolates (N = 347) | Proportionate of Enterobacteriaceae (N = 347) | | | | % of ESBL from total sample size | % of both ESBL and CPE from total isolates | % Both ESBL and MDR from total isolates |
|---|---|---|---|---|---|---|---|---|
| | | ESBL (N (%)) | | Carbapenemase (N (%)) | | | | |
| | | Positive | Negative | Positive | Negative | | | |
| Hospital cafeteria (N = 75) | *E. coli* | 13 (26.5) | 36 (73.5) | 2 (4.1) | 47 (95.9) | 13 (4.5) | 2 (0.58) | 13 (3.7) |
| | *K. pneumoniae* | 5 (29.4) | 12 (70.6) | 3 (17.6) | 14 (82.4) | 5 (1.7) | 3 (0.86) | 5 (1.44) |
| | *C. freundii* | 2 (50) | 2 50) | 0 | 4 (100) | 2 (0.7) | - | 2 (0.57) |
| | *P. mirabilis* | 1 (50) | 1 (50) | 0 | 2 (100) | 1 (0.35) | - | 1 (0.288) |
| | Others | 0 | 3 (100) | 0 | 3 (100) | - | - | - |
| | Total | 21 (28) | 54 (72) | 5 (6.7) | 70 (93.3) | 21 (7.2) | 5 (1.44) | 21 (6.05) |
| CHMS campus cafeteria (N = 55) | *E. coli* | 9 (22.5) | 31 (77.5) | 1 (2.5) | 39 (97.5) | 9 (3.1) | 1 (0.288) | 9 (2.6) |
| | *K. pneumoniae* | 7 (63.6) | 4 (36.7) | 0 | 11 (100) | 7 (2.4) | - | 7 (2) |
| | *C. freundii* | 0 | 2 (100) | 0 | 2 (100) | - | - | - |
| | *P. mirabilis* | - | - | - | - | - | - | - |
| | Others* | 0 | 2 (100) | 0 | 2 (100) | - | - | - |
| | Total | 16 (29.1) | 39 (70.9) | 1 (1.8) | 54 (98.2) | 16 (5.5) | 1(0.288) | 16 (4.6) |
| Maraki campus Cafeteria (N = 105) | *E. coli* | 9 (11.7) | 68 (88.3) | 1 (1.3) | 76 (98.7) | 9 (3.1) | 1 (0.288) | 9 (2.6) |
| | *K. pneumoniae* | 3 (15) | 22 (85) | 0 | 25 (100) | 3 (1.03) | - | 3 (0.86) |
| | *C. freundii* | 0 | 1 (100) | 0 | 1 (100) | - | - | - |
| | *P. mirabilis* | - | - | - | - | - | - | - |
| | Others * | - | 2 (100) | - | 2 (100) | - | - | - |
| | Total | 12 (11.4) | 93 (88.6) | 1 (0.95) | 104 (99.05) | 12 (4.13) | 1 (0.288) | 12 (3.45) |
| Atse Tewodros campus cafeteria (N = 68) | *E. coli* | 6 (12.5) | 42 (87.5) | 0 | 48 (100) | 6 (2.06) | - | 6 (1.72) |
| | *K. pneumoniae* | 1 (9.1) | 10 (90.9) | 0 | 11 (100) | 1 (0.35) | - | 1 (0.288) |
| | *C. freundii* | 0 | 5 (100) | 0 | 5 (100) | - | - | - |
| | *P. mirabilis* | - | - | - | - | - | - | - |
| | Others* | 0 | 4 (100) | 0 | 4 (100) | - | - | - |
| | Total | 7 (10.3) | 61 (89.7) | 0 | 68 (100) | 7 (2.4) | - | 7 (2) |
| Atse Fasile campus cafeteria (N = 24) | *E. coli* | 4 (28.6) | 10 (71.4) | 0 | 14 (100) | 4 (1.4)) | - | 4 (1.15) |
| | *K. pneumoniae* | 1 (25) | 3 (75) | 0 | 4 (100) | 1 (0.35) | - | 1 (0.288) |
| | *C. freundii* | 0 | 1(100) | 0 | 1 (100) | - | - | - |
| | *P. mirabilis* | - | - | - | - | - | - | - |
| | Others* | 0 | 5 (100) | 0 | 5 (100) | - | - | - |
| | Total | 5 (23.8) | 19 (76.2) | 0 | 24 (100) | 5 (1.75) | - | 5 (1.44) |
| Tseda campus cafeteria (N = 20) | *E. coli* | 2 (11.8) | 15 (88.2) | 0 | 17 (100) | 2 (0.7) | - | 2 (0.58) |
| | *K. pneumoniae* | 0 | 2 (100) | 0 | 2 (100) | - | - | - |
| | *C. freundii* | - | - | - | - | - | - | - |
| | *P. mirabilis* | - | - | - | - | - | - | - |
| | Others* | 0 | 1 (100) | 0 | 1 (100) | - | - | - |
| | Total | 2 (10) | 18 (90) | 0 | 20 (6) | 2 (0.7) | - | 2 (0.58) |
| Total (N = 347) | | **63 (18.2)** | **284 (81.8)** | **7 (2)** | **340 (98)** | **63 (21.72)** | **7 (2)** | **63 (18.2)** |

*Others: *M. morganii*, *Providencia* species, *Serratia* species.

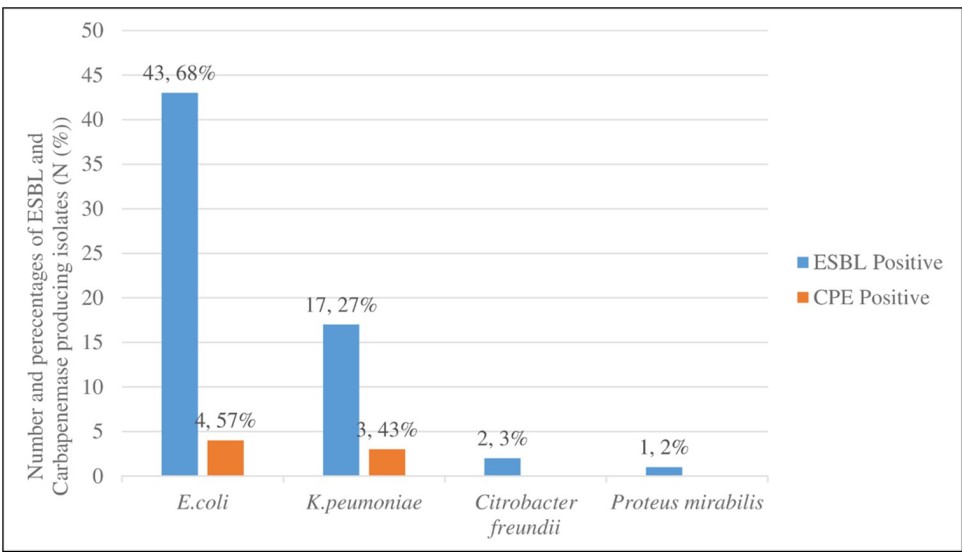

**Fig 2. Proportion of ESBL and Carbapenemase-producing Enterobacteriaceae from the total of ESBL, and Carbapenemase positive isolates at the University of Gondar Comprehensive Specialized Hospital and Students' Cafeteria, Northwest Ethiopia, 2021.**

From the total of ESBL producing isolates leading ESBL producers were *E. coli* 43 (68%) followed by *K. pneumoniae* 17 (27%) similarly the leading Carbapenemase-producing isolates were *E. coli* 4 (57%) followed by *K. pneumoniae* 3 (43%) (Fig 2).

**Antimicrobial resistance patterns of ESBL and Carbapenemase-producing Enterobacteriaceae.** In the current study, cefotaxime, ceftriaxone, ceftazidime, tetracycline, and Sulphamethoxazole-trimethoprim had complete (100% each) resistance found in all of the ESBL-producing isolates. In addition, amoxicillin/clavulanic acid resistance was 55 (87.3%) of these isolates. Resistance to other antibiotics such as Gentamicin, cefoxitin, ciprofloxacin, and chloramphenicol was found to be 29 (46%), 20 (31.7%), 23 (36.5%), and 21 (33.3%) respectively. Only 18 (28.6%) of these ESBL-producing isolates were resistant to Meropenem (Table 6).

All *E. coli* and *K. pneumoniae* ESBL producing isolates were found completely resistant to cefotaxime, ceftriaxone ceftazidime, tetracycline, and Sulphamethoxazole-trimethoprim. This study found that these ESBL producing *E. coli* isolates were also resistant to amoxicillin/clavulanic acid, gentamycin, chloramphenicol, and ciprofloxacin with the rates of 36 (57.1%), 22 (34.9%), 17 (27%), and 13 (20.6%), respectively. Furthermore only 14 (22.2%), and 12 (19%) of these isolates were resistant to cefoxitin and Meropenem respectively. Additionally, *K. pneumoniae* ESBLs producing isolates were also resistant to amoxicillin/clavulanic acid, ciprofloxacin, gentamicin, and chloramphenicol with the rates of 17 (27%), 8 (12.7%), 6 (9.5%), and 4 (6.3%), respectively. In this study, all CPE isolates of *E. coli* and *K. pneumoniae* were resistant to all antibiotics except gentamicin (75%), and (67%) respectively (Table 6).

**Factors associated with fecal carriage rate of ESBL-PE.** After bivariate logistic regression analysis, lack of knowledge on food safety practice (p≤0.001), medical checkup (p≤0.075), history of admission in the last 3 months (p≤0.016), history of diarrhea in the last 3 months (p≤0.000), history of urinary tract infection in the last 3 months (p≤0.003), Antibiotics used in the last 3 months (p≤0.001), History of chronic disease (p≤0.072), drinking unpasteurized milk (p≤0.000), ate raw vegetables (p≤0.015) were included in a multivariate logistic regression analysis. Of socio-demographic factors, lack of knowledge on food safety practice had a

**Table 6. Antimicrobial Resistance patterns of ESBLs and Carbapenemase-producing Enterobacteriaceae among food handlers at the University of Gondar Comprehensive Specialized Hospital and Students' Cafeterias, Northwest Ethiopia, February to June 2021.**

| Antimicrobials | Patterns | ESBL producing isolates (N = 63) | | | | | Carbapenemase-producing isolates (N = 7) | | |
|---|---|---|---|---|---|---|---|---|---|
| | | *E. coli* (N = 43) N (%) | *K. pneumoniae* (N = 17) N (%) | *C. freundii* (N = 2) N (%) | *P. mirabilis* (N = 1) N (%) | Total N (%) | *E. coli* (n = 4) N (%) | *K.pneumoniae* (n = 3) N (%) | Total N (%) |
| AMC | S | 7 (16.3) | 0 | 0 | 1 (100) | 8 (12.7) | 0 | 0 | 0 |
| | R | 36 (83.7) | 17 (100) | 2 (100) | 0 | 55 (87.3) | 4 (100) | 3 (100) | 7 (100) |
| CAF | S | 26 (60.5) | 13 (76.5) | 2 (100) | 1 (100) | 42 (66.7) | 0 | 0 | 0 |
| | R | 17 (39.5) | 4 (23.5) | 0 | 0 | 21 (33.3) | 4 (100) | 3 (100) | 7 (100) |
| CIP | S | 30 (69.8) | 9 (52.9) | 0 | 1(100) | 40 (63.5) | 0 | 0 | 0 |
| | R | 13 (30.2) | 8 (47.1) | 2 (100) | 0 | 23 (36.5) | 4 (100) | 3 (100) | 7 (100) |
| TET | S | 0 | 0 | 0 | 0 | 0 | 0 | 0 | 0 |
| | R | 43 (100) | 17 (100) | 2 (100) | 1 (100) | 63 (100) | 4 (100) | 3 (100) | 7 (100) |
| CAZ | S | 0 | 0 | 0 | 0 | 0 | 0 | 0 | 0 |
| | R | 43 (100) | 17 (100) | 2 (100) | 1 (100) | 63 (100%) | 4(100%) | 3 (100) | 7 (100) |
| CTX | S | 0 | 0 | 0 | 0 | 0 | 0 | 0 | 0 |
| | R | 43 (100) | 17 (100) | 2 (100) | 1 (100) | 63 (100) | 4 (100) | 3 (100) | 7 (100) |
| CRO | S | 0 | 0 | 0 | 0 | 0 | 0 | 0 | 0 |
| | R | 43 (100) | 17 (100) | 2 (100) | 1 (100) | 63 (100) | 4 (100) | 3 (100) | 7 (100) |
| GEN | S | 21 (48.8) | 11 (64.7) | 1 (50) | 1 (100) | 34 (54) | 1 (25) | 1 (33) | 2 (28.5) |
| | R | 22 (51.2) | 6 (35.3) | 1 (50) | 0 | 29 (46) | 3 (75) | 2 (67) | 5 (71.5) |
| SXT | S | 0 | 0 | 0 | 0 | 0 | 0 | 0 | 0 |
| | R | 43 (100) | 17 (100) | 2 (100) | 1 (100) | 63 (100) | 4 (100) | 3 (100) | 7 (100) |
| MER | S | 31 (72.1) | 11 (64.7) | 2 (100) | 1 (100) | 45 (71.4) | 0 | 0 | 0 |
| | R | 12 (27.9) | 6 (35.3) | 0 | 0 | 18 (28.6) | 4 (100) | 3 (100) | 7 (100) |
| CXT | S | 29 (67.4) | 11 (64.7) | 2 (100) | 1 (100) | 43 (68.3) | 0 | 0 | 0 |
| | R | 14 (32.6) | 6 (35.3) | 0 | 0 | 20 (31.7) | 4 (100) | 3 (100) | 7 (100) |

Key: S = Sensitive, R = Resistance, AMC = amoxicillin/clavulanic acid; SXT = Sulphamethoxazole-trimethoprim; TET = tetracycline; CIP = ciprofloxacin;

CHL = chloramphenicol; GEN = gentamycin; CXT = cefoxitin; CRO = ceftriaxone; CTX = cefotaxime, CAZ = ceftazidime; MER = Meropenem.

strong association with the ESBL carriage with P-values 0.003. Of the hygiene-related factors, drinking unpasteurized milk had a strong association with the ESBL carriage with P-values of 0.001. Of the clinical risk factors, the uses of antibiotics in the last 3 months with p-values of 0.016 have demonstrated significant association to the carriage of ESBL-PE (Table 7).

## Discussion

Among 290 study participants, 347 Enterobacteriaceae isolates were identified. Of these, *E. coli* accounts for 245 (70.6%) followed by *K. pneumoniae* 68 (19.6%) and *C. freundii* (3.7%). Our result was in agreement with reports in Egypt *E. coli* [36], and Kuwait *E. coli* [37]. However, this report was lower than a study conducted in Dilla, Ethiopia *E. coli*. This difference may be due to the technique that we used, geographical difference, and population characteristics.

In the present study fecal carriage rate of ESBL- PE among food handlers was 21. 7% (95% CI: 17.2–26.6). This finding is in line with the studies conducted in Dilla, Ethiopia [29]. However, it is lower than the study conducted in Mexico [38], Italy [39], China [40], India [41], Nepal [42], Chad [43], and Egypt [36], and [44]. In contrast, our result is higher than the study conducted in Japan [45, 46], Switzerland [47], Nigeria [48], and Gambia [49]. The methodological and geographical differences, bacteriological media used may contribute to the high

**Table 7. Factors associated with fecal carriage rate of ESBL-PE among food handlers at the University of Gondar Specialized Hospital and students' cafeteria, Northwest Ethiopia, February to June 2021.**

| Variables | Category | Number tested (N = 290) | ESBL Positive (N = 63) N (%) | ESBL Negative (N = 227) N (%) | Bivariate analysis | | Multivariable analysis | |
|---|---|---|---|---|---|---|---|---|
| | | | | | COR(95% CI) | p-value | AOR(95% CI) | p-value |
| Gender | Male | 48 | 7 (14.6) | 41 (85.4) | 1 | | 1 | |
| | Female | 242 | 56 (23.1) | 86 (76.9) | 2.46 (0.84–7.23) | 0.100 | 0.30 (0.08–1.2) | 0.086 |
| Age in years | <20 | 8 | 3 (37.5) | 5 (62.5) | 1 | | 1 | |
| | 20–40 | 240 | 51 (21.3) | 189 (78.7) | 0.60 (0.10–3.61) | 0.577 | | |
| | >40 | 42 | 9 (21.4) | 33 (78.6) | 1.03 (0.43–2.48) | 0.954 | | |
| Level of Education | Illiterate | 57 | 29 (50.9) | 28 (49.1) | 0.16 (0.31–0.97) | 0.020 | 0.7 (0.07–0.16) | 0.081 |
| | Primary | 98 | 13 (13.3) | 85 (86.7) | 1.70 (0.71–1.92) | 0.371 | 0.3 (0.24–0.77) | 0.246 |
| | Secondary and above | 115 | 21 (18.3) | 94 (81.7) | 1 | | 1 | |
| Monthly income | Less than 1000 ETB | 7 | 3 (42.9) | 4 (57.1) | 0.52 (0.08–3.45) | 0.501 | | |
| | Greater than 1500 ETB | 283 | 60 (21.2) | 223 (78.8) | 1 | | | |
| Marital status | Married | 159 | 38 (23.9) | 121 (76.1) | 0.84 (0.44–1.60) | 0.595 | | |
| | Unmarried | 131 | 25 (19.1) | 106 (80.9) | 1 | | | |
| Family size | 1–5 | 262 | 59 (22.5) | 203 (77.5) | 0.59 (0.17–2.05) | 0.407 | | |
| | Greater than 5 | 28 | 4 (14.3) | 24 (85.7) | 1 | | | |
| Service years | 1–5 | 152 | 34 (22.4) | 118 (77.6) | 0.28 (0.38–0.95) | 0.398 | | |
| | Greater than 5 | 138 | 29 (21) | 109 (79) | 1 | | | |
| Knowledge of food safety practice | Yes | 174 | 38 (21.8) | 136 (78.2) | 1 | | | |
| | No | 116 | 25 (21.6) | 91 (78.4) | 3.50 (1.81–6.78) | 0.000 | 1.2 (1.07–1.46) | **0.003*** |
| Medical checkups | Yes | 181 | 18 (9.9) | 163 (90.1) | 1 | | | |
| | No | 109 | 45 (41.3) | 64 (58.7) | 0.52 (.26–1.0) | 0.075 | | |
| History of admission in the last 3 months | Yes | 6 | 6 (100) | 0 | 0.21 (0.06–0.75) | 0.016 | 2.1 (0.47–9.16) | 0.331 |
| | No | 284 | 57 (20) | 227 (80) | 1 | | 1 | |
| History of medical instrumentation | Yes | 8 | 6 (75) | 2 (25) | 1 | 0.522 | 1 | |
| | No | 282 | 57 (20.2) | 225 (79.8) | 0.587 (0.11–3.0) | | 0.9 (0.18–5.24) | 0.970 |
| History of diarrhea in the last 3 months | Yes | 93 | 35 (37.6) | 58 (62.4) | 0.28 (0.14–0.53) | 0.000 | 1.5 (0.62–3.67) | 0.371 |
| | No | 197 | 28 (14.2) | 169 (85.8) | 1 | | 1 | |
| History of UTI | Yes | 58 | 22 (37.9) | 36 (62.1) | 0.35 (0.18–0.69) | 0.003 | 1.4 (0.56–3.38) | 0.510 |
| | No | 232 | 41 (17.7) | 191 (82.3) | 1 | | 1 | |
| Antibiotics used in the last 3 months | Yes | 129 | 45 (34.9) | 84 (65.1) | 0.19 (0.09–0.39) | 0.000 | 4.14 (1.31–13) | **0.016*** |
| | No | 161 | 18 (11.2) | 143 (88.8) | 1 | | 1 | |

(*Continued*)

**Table 7.** (Continued)

| Variables | Category | Number tested (N = 290) | ESBL Positive (N = 63) N (%) | ESBL Negative (N = 227) N (%) | Bivariate analysis | | Multivariable analysis | |
|---|---|---|---|---|---|---|---|---|
| | | | | | COR(95% CI) | p-value | AOR(95% CI) | p-value |
| History of chronic disease | Yes | 38 | 13 (34.2) | 25 (65.8) | 0.48(0.21–1.07) | 0.072 | 0.81 (0.31–2.1) | 0.668 |
| | No | 252 | 50 (19.8) | 202 (80.2) | 1 | | 1 | |
| Hand washing habit | Yes | 264 | 60 (22.7) | 204 (77.3) | 1 | | | |
| | No | 26 | 3 (11.5) | 23 (84.5) | 0.39 (0.09–1.72) | 0.215 | | |
| Fingernail status | trimmed | 273 | 58 (21.2) | 215 (78.8) | 1 | | 1 | 0.566 |
| | Not trimmed | 17 | 5 (29.4) | 12 (70.6) | 1.59 (0.49–5.13) | 0.433 | 0.69 (0.19–2.49) | |
| Wear hair garment | Yes | 259 | 58 (22.4) | 201 (77.6) | 1 | | 1 | 0.795 |
| | No | 31 | 5 (16.1) | 26 (83.9) | 1.0 (0.36–2.78) | 0.993 | 0.86 (0.27–2.74) | |
| Drinking unpasteurized milk | Yes | 71 | 35 (49.3) | 36 (50.7) | 8.9 (4.48–17.71) | 0.000 | 10.4 (3.8–28.8) | **0.001**[*] |
| | No | 219 | 27 (12.3) | 192 (87.7) | 1 | | 1 | |
| Eating raw meat | Yes | 59 | 17 (28.8) | 42 (71.2) | 1.58 (0.77–3.23) | 0.211 | 0.60 (0.24–151) | 0.282 |
| | No | 231 | 46 (19.9) | 185 (80.1) | 1 | | 1 | |
| Eating raw vegetables | Yes | 148 | 6 (4.1) | 142 (95.9) | 0.44 (0.22–0.85) | 0.015 | 0.4 (0.15–1.26) | 0.125 |
| | No | 142 | 57 (40.1) | 85 (59.9) | 1 | | 1 | |

Abbreviations, COR = crude odds ratio, AOR = adjusted odds ratio, CI = confidence interval.

magnitude of ESBL production. These affected food handlers pose a hazard to the spread of MDR gram-negative bacteria in the schools, in their homes, and communities where they prepare and serve food. As demonstrated in a cohort study of food handlers in Japan, if not treated, some of these ESBL carriers could function as long-term reservoirs for the spread of these pathogens [45].

In the current study, *E. coli* (68%) and *K. pneumoniae* (27%) were the most common ESBL producing isolates of Enterobacteriaceae. *E. coli* is a leading cause of ESBL producing isolates. This result is agreed with the report in Nepal [42], China [40], Chad [43], Egypt [36], and Dilla, Ethiopia [29], where *E. coli* was the predominant ESBL producing Enterobacteriaceae than *K. pneumoniae*. Additionally, *C. freundii* isolates were the third ESBL producing isolates with a prevalence of (3%). This finding was in line with the study conducted in Nepal [42]. However, *K. pneumoniae* as a common ESBL producer was reported in the Gambia *E. coli* and *K. pneumoniae* [49] and Bahir Dar, Ethiopia *E. coli* and *K. pneumoniae* [50]. These bacteria cause the resistance genes to be transmitted to other strains of *E. coli* and *K. pneumoniae* in the gastrointestinal tract, which can be fatal. Furthermore, when those carriers are admitted to hospitals, they can easily spread the infection to other patients. An increased overall drug resistance rate against different categories of drugs could be explained by rapid adaptation of those strains to the harsh environment, up-regulation of intrinsic resistance mechanisms, and rapid acquisition and transmission of drug resistance genes through mobile genetic elements [51].

In this study, the overall prevalence of CPE was 7 (2%) (95%CI = 0.7–4.5), this finding is concordant with other studies such as in the Gambia, [49], Egypt ([44], and China [40]).

However, it is lower than studies in Kuwaiti [37, 52]. This discrepancy of isolation may be due to geographical location, population characteristics, poor sanitary practice, incorrect use of antibiotics, cross-border of the population with other high-prevalence countries, sample size, and methodological heterogeneity could all contribute to variations in CPE prevalence.

In this study, the overall prevalence of MDR-E was 147 (42.3%: 95% CI: 39.7–51). This study is consistent with the study done in Egypt [53]. But it was higher than a study conducted in, Kenya [54], Kuwaiti [55], and Qatar [56]. This discrepancy could be due to irrational uses of antibiotics, poor personal and environmental hygiene in the study area, a lack of proper diagnostic tools, consumption of animal products that use antibiotics for growth promotion and treatment, an increase in MDR strains over time, differences in the study population and failure of patient adherence to their medication [57].

Meropenem was found to be the most effective antibiotic for Enterobacteriaceae isolates (sensitivity 95%) which was comparable with the study done in Nigeria [58], and Turkey [59]. In the present study isolates were resistant to tetracycline (61.1%), and amoxicillin/clavulanic acid (57.3%) which was lower than the study conducted in Nigeria amoxicillin/clavulanic acid, and tetracycline [48].

Regarding antimicrobial resistance rate, all ESBL-producing Enterobacteriaceae were resistant to cefotaxime, ceftazidime, and ceftriaxone. This finding agrees with studies conducted in the Gambia [49], Nigeria [48], and Dilla, Ethiopia [29]. This indicates ESBL-producing Enterobacteriaceae were rapidly emerging in developing countries. In this research, ESBL-producing Enterobacteriaceae are found to be not only completely resistant to third-generation cephalosporin but also other non-beta-lactam antibiotics such tetracycline and Sulphamethoxazole-trimethoprim. This finding concords with the study conducted in Nigeria [58], and Dilla, Ethiopia [29]. Additionally, the finding of this study showed that Meropenem had a better performance against ESBL-producing Enterobacteriaceae than other antibiotics including the cephalosporin group. However, our result is different from the study conducted in Nigeria [48].

In this study all ESBL producing isolates were MDR. The MDR nature of ESB-PE may be explained by the fact that they are plasmid-mediated enzymes that carry multi-resistant genes through plasmid, transposon, and integron, and can easily be transferred to other bacteria by conjugation, transduction, or transformation [60].

High fecal colonization of ESBL and Carbapenemase-producing Enterobacteriaceae was attributed to the contribution of different risk factors. The risk factors mainly identified from this study after doing multivariable analysis were lack of knowledge in food safety practices, drinking unpasteurized milk, and history of antibiotics usage in the last 3 months. This result showed that the fecal carriage of ESBLs in food-handlers was significantly associated with, drinking unpasteurized milk, history of antibiotics usage in the last 3 months, and lack of knowledge in food safety practices as found in other studies from China [40], Gambia [49], and Dilla, Ethiopia [29]. However, this finding is inconsistent with the study conducted in Japan reported that history of antibiotic usage had no significant association with the fecal carriages of ESBL-positive Enterobacteriaceae [46].

The extended-spectrum beta-lactamase carriage rate was highest among food handlers with a history of antibiotics usage in the last 3 months compared to those who did not have used antibiotics. The history of antibiotic use in the last 3 months was more than 4 folds higher (AOR, 4.14 (95% CI, 1.3–13)), compared to those who did not have used antibiotics This result is consistent with the reports of the study conducted from China [40], Gambia [49], and Dilla, Ethiopia [29]. This can be explained by the fact that the usage of antibiotics in the past and present can provide a significant competitive advantage for the development of resistant bacteria and the spread of ESBL-producing bacteria in the population. Antibiotics are taken without

a prescription in most developing countries, such as Ethiopia, which can lead to antibiotic overuse or misuse, which triggers the emergence and spread of antimicrobial-resistant bacteria [61]. Furthermore, poor personal hygiene habits in developing countries may increase the prevalence of bacterial gastrointestinal infections, which may contribute to the rise in antibiotic resistance. This raises the possibility of treatment failure, which could have profound consequences. This can be avoided by conducting a bacterial culture, and antimicrobial susceptibility test, and adhering to proper antibiotic stewardship whenever treating patients with suspected gastrointestinal infections.

### Limitation of the study

The present study has the following limitations:

Antibiotic resistance encoding genes (ARGs) of isolates were not detected as a confirmatory test due to a lack of molecular techniques and primers. Because the study was cross-sectional, there was no follow-up to treat ESBL carriers and identify how they could infection remain.

### Conclusions and recommendations

In this study, the prevalence of fecal carriage of MDR Enterobacteriaceae, ESBL-PE, and Carbapenemase-producing Enterobacteriaceae among food-handlers were high and this is a threat to the public. The most frequent ESBL-producing Enterobacteriaceae were *E. coli* and *K. pneumoniae*. In this study, Meropenem and cefoxitin were the most effective antimicrobial agents compared to other tested antibiotics. The overall MDR prevalence in this study was 147 (42.3%). Accordingly, from the isolates of *C. freundii* were 9 (69.2%) MDR followed by *P. mirabilis* 1 (50%), *E. coli* 104 (42.4%), and *K. pneumoniae* 28 (41.2%). Antibiotics usage in the last 3 months and drinking unpasteurized milk were associated with the carriage of the ESBL-PE. The result indicates that ESBL and CPE carriage is widespread calling for mass screening of the community for ESBL and CPE. The emergence of ESBL and Carbapenemase-producing Enterobacteriaceae necessitates the implementation of strict infection prevention and control strategies, surveillance of antibiotic resistance, and following proper antibiotic stewardship, which plays an important role in clinical decision-making. The level of evidence is still low and requires further research by incorporating research on molecular confirmation and profiling of the ESBL and Carbapenemase-producing isolates to identify resistance genes.

### Supporting information

**S1 File.**
(RTF)

**S1 Protocol. The AST interpretation chart (extracted from CLSI, guideline 2020).** Based on CLSI criteria, interpret the zones sizes of each antimicrobial, reporting the organism as 'Resistant', 'Intermediate/Moderately susceptible', 'Susceptible see below the table'.
(RTF)

**S2 Protocol. Lab protocol.**
(RTF)

### Acknowledgments

The authors would like to thank the Department of Medical Microbiology, School of Biomedical and Laboratory Sciences, College of Medicine and Health Sciences, the University of Gondar and we also acknowledge study participants.

## Author Contributions

**Conceptualization:** Azanaw Amare, Feleke Moges.

**Data curation:** Azanaw Amare.

**Formal analysis:** Azanaw Amare.

**Investigation:** Azanaw Amare.

**Project administration:** Azanaw Amare.

**Software:** Azanaw Amare.

**Supervision:** Setegn Eshetie, Desie Kasew, Feleke Moges.

**Writing – original draft:** Azanaw Amare.

**Writing – review & editing:** Azanaw Amare, Setegn Eshetie, Desie Kasew, Feleke Moges.

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
