## [Decision Letter · Decision Letter 0]

10 Jan 2022

PONE-D-21-38463Fecal carriage rate of Extended-spectrum beta-lactamase and carbapenemase-producing Enterobacteriaceae among healthy food handlers at the University of Gondar Cafeterias, Northwest EthiopiaPLOS ONE

Dear Dr. Muche,

Thank you for submitting your manuscript to PLOS ONE. After careful consideration, we feel that it has merit but does not fully meet PLOS ONE’s publication criteria as it currently stands. Therefore, we invite you to submit a revised version of the manuscript that addresses the points raised during the review process.

ACADEMIC EDITOR: Please revise the manuscript according to the reviewer comments. A major revision is required.==============================

We look forward to receiving your revised manuscript.

Kind regards,

Abdelazeem Mohamed Algammal, Prof, Ph.D

Academic Editor

PLOS ONE

Journal Requirements:

4. Please amend the manuscript submission data (via Edit Submission) to include authors Setegn Eshetie, Desie Kasaw, and Feleke Moges.

Reviewers' comments:

Reviewer's Responses to Questions

**Comments to the Author**

1. Is the manuscript technically sound, and do the data support the conclusions?

Reviewer #1: Partly

Reviewer #2: Partly

2. Has the statistical analysis been performed appropriately and rigorously? 

Reviewer #1: Yes

Reviewer #2: Yes

3. Have the authors made all data underlying the findings in their manuscript fully available?

Reviewer #1: No

Reviewer #2: Yes

4. Is the manuscript presented in an intelligible fashion and written in standard English?

Reviewer #1: No

Reviewer #2: No

5. Review Comments to the Author

Reviewer #1: Comments to authors:

- The current study has a significant impact, but it needs a major revision:

- The manuscript should be revised for grammar mistakes.

- Please write the scientific names of bacterial pathogens and genes in the correct form all over the manuscript and in the References section (should be italic).

-The title is broad, please modify the title.

- Add more details about the used methods and most prevalent results in the abstract.

-In the introduction: discuss the public health importance of E. coli, and K. pneumonia and their virulence determinants.

-Improve the aim of work.

Methods:

-Explain the methods of isolation and identification in detail??

-Specific references should be added to all the used methods and techniques.

- Antimicrobial susceptibility testing: Add the manufacturing company, city, and country for the used reagents and antimicrobial discs.

-PCR based detection of virulence genes and antimicrobial resistance genes in the most prevalent retrieved bacterial species should be carried out if applicable (or addresses this point in the study limitations)

--Results:

- Discuss in detail the phenotypic characters of the recovered isolates.

-increase the resolution of different Figures: Please improve.

-PCR based detection of virulence genes and antimicrobial resistance genes in the most prevalent retrieved bacterial species should be carried out if applicable (or addresses this point in the study limitations)

-The correlation between the phenotypic and genotypic MDR should be performed.

-Discussion:

- Please improve.

-Please improve the main conclusion of the manuscript.

Reviewer #2: -The current study is interesting; however, the authors should address the following comments to improve the quality of the manuscript:

Title:

I think the work would benefit from the title that contains the main conclusion of the study (should be derived from the conclusion). Please modify the title.

Abstract:

- The abstract must illustrate the used methods and the most prevalent results (give more hints about methods and results). Besides, rephrase the aim of the work and the main conclusion of your findings.

Introduction: (it needs to be more informative)

-Give a hint about the virulence factors, different infections caused by of E. coli and K. pneumonia, and the mechanism of disease occurrence.

- The authors should illustrate the public health importance concerning the emergence of multidrug-resistant (MDR) bacterial pathogens that reflect the necessity of new potent and safe antimicrobial agents. Several studies proved the widespread MDR- bacterial pathogens;

Authors could add the following paragraph:

Multidrug resistance has been increased all over the world that is considered a public health threat. Several recent investigations reported the emergence of multidrug-resistant bacterial pathogens from different origins including humans, birds, cattle, and fish that increase the need for routine application of the antimicrobial susceptibility testing to detect the antibiotic of choice as well as the screening of the emerging MDR strains. You should cite the following valuable studies:

1.PMID: 33177849

2.PMID: 32994450

3.PMID: 33061472

4.PMID: 33947875

5.PMID: 34445951

6.PMID: 33188216

7.https://doi.org/10.1016/j.aquaculture.2021.737643

8.PMID: 30150182

-Rephrase the aim of the work to be clear and better sound.

Material and methods: Illustrate your methods with subtitles:

-Add this subtitle: Bacterial Isolation and identification:

•Discuss in detail the methods of isolation and identification of E. coli, K. pneumonia, and other members of Enterobacteriaceae. Besides, specific references should be added.

•Add the company, city, and country of the used bacterial media and reagents that were used in the biochemical identification of isolates. Also, enumerate all used biochemical reactions.

- Antimicrobial susceptibility testing:

•Illustrate the antimicrobial classes of the tested antimicrobial agents within the text.

•The authors are advised to classify the tested isolates to MDR , XDR, and PDR as described by Magiorakos et al.

Magiorakos AP, Srinivasan A, Carey RB, Carmeli Y, Falagas ME, Giske CG, et al. Multidrug-resistant, extensively drug-resistant and pandrug-resistant bacteria: An international expert proposal for interim standard definitions for acquired resistance. Clin Microbiol Infect. 2012; 18:268–81. doi:10.1111/j.1469-0691.2011.03570.x.

- Why did you ignore the detection of antibiotic resistance genes in the recovered isolates??

•Please use PCR to detect antibiotic resistance genes, followed by gene sequencing. Afterward, the correlation between phenotypic and genotypic multidrug resistance should be performed.

-Add more details about the software used in the statistical analyses.

-Results:

-Add this subtitle: Phenotypic characteristics of the recovered isolates.

•Illustrate in detail the phenotypic characteristics of the recovered isolates, especially E. coli and K. pneumonia,

-Antimicrobial susceptibility testing:

•-Illustrate in a new table the occurrence of MDR (Multidrug resistance) among the recovered isolates (illustrate the names of the antimicrobial classes and different antibiotics):

No. of strains%Type of resistance

R, MDR, and XDRPhenotypic multidrug resistance

(Antimicrobial classes and different antibiotics).The antibiotic -resistance genes

- Increase the resolution of all figures (it should be 600 dpi).

-Discussion:

- The authors are advised to illustrate the real impact of their findings without repetition of results.

-Illustrate the different mechanisms of antimicrobial resistance in E. coli and K. pneumonia.

-Conclusion

- Should be rephrased to be sounded. A real conclusion should focus on the question or claim you articulated in your study, which resolution has been the main objective of your paper?

6. PLOS authors have the option to publish the peer review history of their article (what does this mean?). If published, this will include your full peer review and any attached files.

Reviewer #1: No

Reviewer #2: No

---

## [Author Response · Author response to Decision Letter 0]

17 Feb 2022

Date: 17/02/2022

Response to Reviewers

To reviewers and journal editors:

Dear reviewers,

We appreciate for spending your precious time and forwarding your valuable comments, which have significantly improved our application. We are also grateful for this positive feedback. Please see below, bold, for a point-by-point response to the reviewers. All page numbers refer to the revised manuscript file with tracked changes. All modifications in the manuscript have been highlighted in red. 

We are looking forward to hearing from you in due course,

Sincerely, 

Azanaw Amare (Corresponding author)

Azanaw03@gmail.com

Title of the manuscript: “High prevalence of fecal carriage of Extended-spectrum beta-lactamase and Carbapenemase-producing Enterobacteriaceae among food handlers at the University of Gondar, Northwest Ethiopia.” 

Manuscript Number: PONE-D-21-38463

Below we provide the point-by-point responses. All modifications in the manuscript have been highlighted in red.

Thank you for submitting your manuscript to PLOS ONE. After careful consideration, we feel that it has merit but does not fully meet PLOS ONE’s publication criteria as it currently stands. Therefore, we invite you to submit a revised version of the manuscript that addresses the points raised during the review process.

ACADEMIC EDITOR: Please revise the manuscript according to the reviewer comments. A ma-jor revision is required.

• A rebuttal letter that responds to each point raised by the academic editor and review-er(s). You should upload this letter as a separate file labeled 'Response to Reviewers'.

• A marked-up copy of your manuscript that highlights changes made to the original ver-sion. You should upload this as a separate file labeled 'Revised Manuscript with Track Changes'.

If applicable, we recommend that you deposit your laboratory protocols in protocols.io to en-hance the reproducibility of your results. Protocols.io assigns your protocol its own identifier (DOI) so that it can be cited independently in the future. For instructions see: https://journals.plos.org/plosone/s/submission-guidelines#loc-laboratory-protocols. Additionally, PLOS ONE offers an option for publishing peer-reviewed Lab Protocol articles, which describe protocols hosted on protocols.io. Read more information on sharing protocols at https://plos.org/protocols?utm_medium=editorial-email&utm_source=authorletters&utm_campaign=protocols.

We look forward to receiving your revised manuscript.

Kind regards,

Abdelazeem Mohamed Algammal, Prof, Ph.D

Academic Editor

PLOS ONE

Authors’ response: we included the DOI link in the Methods section of the manuscript 

Journal Requirements:

 Response: thank you very much. We believed that we are following PLOS ONE’s style requirements.

Response: thank you very much. We accept and correct it accordingly. 

3. In your Data Availability statement, you have not specified where the minimal data set under-lying the results described in your manuscript can be found. PLOS defines a study's minimal data set as the underlying data used to reach the conclusions drawn in the manuscript and any addi-tional data required to replicate the reported study findings in their entirety. All PLOS journals require that the minimal data set be made fully available. For more information about our data policy, please see http://journals.plos.org/plosone/s/data-availability.

"Upon re-submitting your revised manuscript, please upload your study’s minimal underlying da-ta set as either Supporting Information files or to a stable, public repository and include the rele-vant URLs, DOIs, or accession numbers within your revised cover letter. For a list of acceptable repositories, please see http://journals.plos.org/plosone/s/data-availability#loc-recommended-repositories. Any potentially identifying patient information must be fully anonymized.

Important: If there are ethical or legal restrictions to sharing your data publicly, please explain these restrictions in detail. Please see our guidelines for more information on what we consider unacceptable restrictions to publicly sharing data: http://journals.plos.org/plosone/s/data-availability#loc-unacceptable-data-access-restrictions. Note that it is not acceptable for the au-thors to be the sole named individuals responsible for ensuring data access.

Response: thank you. Please update it 

4. Please amend the manuscript submission data (via Edit Submission) to include authors Setegn Eshetie, Desie Kasaw, and Feleke Moges.

 Response: I amend it and I notify during the revision 

Reviewers' comments:

Reviewer's Responses to Questions

Comments to the Author

1. Is the manuscript technically sound, and do the data support the conclusions?

The manuscript must describe a technically sound piece of scientific research with data that sup-ports the conclusions. Experiments must have been conducted rigorously, with appropriate con-trols, replication, and sample sizes. The conclusions must be drawn appropriately based on the data presented. 

Reviewer #1: Partly

Reviewer #2: Partly

2. Has the statistical analysis been performed appropriately and rigorously? 

Reviewer #1: Yes

Reviewer #2: Yes

3. Have the authors made all data underlying the findings in their manuscript fully available?

The PLOS Data policy requires authors to make all data underlying the findings described in their manuscript fully available without restriction, with rare exceptions (please refer to the Data Availability Statement in the manuscript PDF file). The data should be provided as part of the manuscript or its supporting information, or deposited to a public repository. For example, in ad-dition to summary statistics, the data points behind means, medians, and variance measures should be available. If there are restrictions on publicly sharing data—e.g. participant privacy or use of data from a third party—those must be specified.

Reviewer #1: No

Reviewer #2: Yes

4. Is the manuscript presented in an intelligible fashion and written in Standard English?

Reviewer #1: No

Reviewer #2: No

5. Review Comments to the Author

Please use the space provided to explain your answers to the questions above. You may also in-clude additional comments for the author, including concerns about dual publication, research ethics, or publication ethics. (Please upload your review as an attachment if it exceeds 20,000 characters)

Reviewer #1: Comments to authors:

comments: The current study has a significant impact, but it needs a major revision:

Response: thank you very much for your positive feedback

Comments: The manuscript should be revised for grammar mistakes.

Response: We are grateful for the suggestion and thank you. We went throughout the entire manuscript to eliminate grammatical and editing mistakes.

Comments: Please write the scientific names of bacterial pathogens and genes in the correct form all over the manuscript and in the References section (should be italic).

Response: We are also grateful for the suggestion and thank you. We went through the en-tire manuscript to italic bacterial pathogens and genes.

Comments: The title is broad, please modify the title.

Response: Thank you very much. We agree with this comment. Hence, we have amended the title as follows: “High prevalence of fecal carriage of Extended-spectrum beta-lactamase and Carbapenemase-producing Enterobacteriaceae among food handlers at the University of Gondar, Northwest Ethiopia.” 

Comments: Add more details about the used methods and most prevalent results in the abstract.

Responses: We agree with this and have incorporated your comments 

Comments: In the introduction: discuss the public health importance of E. coli, and K. pneumo-nia and their virulence determinants.

Response: : Thank you for this suggestion. It would have been interesting to explore this as-pect. We tried to add the following suggestion on page 3 (Line 71&72, 77-87).

Comments: Improve the aim of the work.

Response: Thank you very much for the reminder. We have made revisions accordingly as follows, “Therefore, this study aimed to assess fecal carriage rate, associated factors, and antimicrobial resistance patterns of Extended-spectrum Beta-lactamase and Car-bapenemase-producing Enterobacteriaceae among food handlers at the University of Gon-dar Cafeteria.” Page 4 of the revised manuscript.

Methods:

Comments: Explain the methods of isolation and identification in detail??. Specific references should be added to all the used methods and techniques

Response: We agree with the reviewer’s assessment. Accordingly, we have revised this in the method section and added the suggested comments as follows (Page 6; and line 154-170).

 Comments: Antimicrobial susceptibility testing: Add the manufacturing company, city, and country for the used reagents and antimicrobial discs.

Response: Thank you very much for this reminder. We have made revisions on method parts accordingly.

Comments: PCR based detection of virulence genes and antimicrobial resistance genes in the most prevalent retrieved bacterial species should be carried out if applicable (or addresses this point in the study limitations)

Response: We agree that this is a potential limitation of the study. We have added this as a limitation on pages 27 and 28 of the revised manuscript. The revised sentence is as follows (Line 445-449). “Antibiotic resistance encoding genes (ARGs) of isolates were not detected as a confirmatory test due to a lack of molecular techniques and primers”.

--Results: 

Comments: Discuss in detail the phenotypic characters of the recovered isolates.

Response: Thank you. The general phenotypic characteristics of the recovered isolates espe-cially for E. coli and K. pneumoniae are described in the result sections on page 11, (lines 264-280).

Comments:-increase the resolution of different Figures: Please improve.

Response: thank you very much. We tried to increase the resolution using the PACE digital diagnostic tool.

Comments:-PCR based detection of virulence genes and antimicrobial resistance genes in the most prevalent retrieved bacterial species should be carried out if applicable (or addresses this point in the study limitations)

Response: We agree that this is a potential limitation of the study. We have added this as a limitation on pages 27 and 28 of the revised manuscript. The revised sentence is as follows (Line 445-449). “Antibiotic resistance encoding genes (ARGs) of isolates were not detected as a confirmatory test due to a lack of molecular techniques and primers.”

Comments:-The correlation between the phenotypic and genotypic MDR should be performed.

Response: We agree that this is a potential limitation of the study. We have added this as a limitation on pages 27 and 28 of the revised manuscript. The revised sentence is as follows (Line 445-449). “Antibiotic resistance encoding genes (ARGs) of isolates were not detected as a confirmatory test due to a lack of molecular techniques and primers”.

-Discussion: 

Comments: Please improve.

Response: Thanks for your kind reminders. We revised the discussion section (page 25 and 26, lines 364-417).

Comments: Please improve the main conclusion of the manuscript.

Response: We revised according to the amended title of the manuscript

Reviewer #2: 

Comments: The current study is interesting; however, the authors should address the following comments to improve the quality of the manuscript:

Response: Thank you very much for this positive feedback

Title: 

Comments: I think the work would benefit from the title that contains the main conclusion of the study (should be derived from the conclusion). Please modify the title.

Response: Thank you very much for pointing out this suggestion. We agree with this com-ment and suggestion. Therefore, we have amended the title as follows: “High prevalence of fecal carriage of Extended-spectrum beta-lactamase and Carbapenemase-producing Enter-obacteriaceae among food handlers at the University of Gondar, Northwest Ethiopia.”

Abstract:

 Comments: The abstract must illustrate the used methods and the most prevalent results (give more hints about methods and results). Besides, rephrase the aim of the work and the main con-clusion of your findings.

Response: We agree with this and have incorporated your suggestions and comments (page 2, line 36-38, 40&41; 43-45, 47-52, 59-61, 62-64).

Comments: Introduction: (it needs to be more informative)

Response: thank you very much we tried and revised it to be more informative. 

Comments: Give a hint about the virulence factors, different infections caused by E. coli and K. pneumonia, and the mechanism of disease occurrence.

Response: Thank you for this suggestion. It would have been interesting to explore this as-pect. We tried to add the following suggestion on page 3 (Line 71&72, 77-87).

Comments: The authors should illustrate the public health importance concerning the emergence of multidrug-resistant (MDR) bacterial pathogens that reflect the necessity of new potent and safe antimicrobial agents. Several studies proved the widespread MDR- bacterial pathogens;

Responses: Agree. We have made the following changes on page 4 (lines 88-92).

Comments: Authors could add the following paragraph:

Multidrug resistance has been increased all over the world that is considered a public health threat. Several recent investigations reported the emergence of multidrug-resistant bacterial path-ogens from different origins including humans, birds, cattle, and fish that increase the need for routine application of the antimicrobial susceptibility testing to detect the antibiotic of choice as well as the screening of the emerging MDR strains. You should cite the following valuable stud-ies:

1. PMID: 33177849

2. PMID: 32994450

3. PMID: 33061472

4. PMID: 33947875

5. PMID: 34445951

6. PMID: 33188216

7.https://doi.org/10.1016/j.aquaculture.2021.737643

8. PMID: 30150182

Response: thank you very much for your direction, suggestion, and comments. We agree with this and have incorporated your suggestion in the revised manuscript on page 4 (line 88-92).

Comments: Rephrase the aim of the work to be clear and better sound.

Response: Thank you very much for pointing this out. We revised the objective as follows: “Therefore, this study aimed to assess fecal carriage rate, associated factors, and antimicro-bial resistance patterns of Extended-spectrum Beta-lactamase and Carbapenemase-producing Enterobacteriaceae among food handlers at the University of Gondar Cafete-ria.” Page 4 of the revised manuscript

Comments: Material and methods: Illustrate your methods with subtitles:

Response: thank you very much. We incorporated this as the subtitle

Comments: Add this subtitle: Bacterial Isolation and identification:

Response: thank very much, we incorporate this as the subtitle 

Comments: Discuss in detail the methods of isolation and identification of E. coli, K. pneumonia, and other members of Enterobacteriaceae. Besides, specific references should be added.

Response:

Comments: Add the company, city, and country of the used bacterial media and reagents that were used in the biochemical identification of isolates. Also, enumerate all used biochemical reac-tions.

Response: We inserted the suggested comments accordingly

- Antimicrobial susceptibility testing:

Comments: Illustrate the antimicrobial classes of the tested antimicrobial agents within the text.

Response: Thank you for this suggestion. We inserted it with text on page 7, (line 178-183)

Comments: The authors are advised to classify isolated isolates to MDR, XDR, and PDR as de-scribed by Magiorakos et al.

Magiorakos AP, Srinivasan A, Carey RB, Carmeli Y, Falagas ME, Giske CG, et al. Multidrug-resistant, extensively drug-resistant and pan drug-resistant bacteria: An international expert pro-posal for interim standard definitions for acquired resistance. Clin Microbiol Infect. 2012; 18:268–81. doi:10.1111/j.1469-0691.2011.03570.x.

Response: Thank you for this suggestion. The antimicrobial classes of the tested antimicro-bial agents are indicated in Table 2 for each type of isolates, while the MDR, is presented in table 3 page 14. But for testing XDR, and PDR, we agree with potential limitations because we did not test all antimicrobial from the antimicrobial class of the drugs.

Comments: Why did you ignore the detection of antibiotic resistance genes in the recovered iso-lates??

Response: We agree that this is a potential limitation of the study. We have added this as a limitation and stated as: “Antibiotic resistance encoding genes (ARGs) of isolates were not detected as a confirmatory test due to a lack of molecular techniques and primers.”

Comments: Please use PCR to detect antibiotic resistance genes, followed by gene sequencing. Afterward, the correlation between phenotypic and genotypic multidrug resistance should be performed.

Response: We agree that this is a potential limitation of the study. We have added this as a limitation on page 27 and 28 of the revised manuscript. The revised sentence is as follows (Line 345-349). “Because of the lack of molecular methods and primes, antibiotic resistance encoding genes (ARGs) of isolates were not detected as a confirmatory test”.

Comments: Add more details about the software used in the statistical analyses.

Response: We think this is an excellent suggestion. However, we believe that these soft wares used for statistical analyses (Epi-Data version 4.6 for data entry and SPSS version 26 for statistical analysis) are open access soft wares everybody can access them. The instructions for use are freely available. Therefore, describing these statistical tools beyond the level of this detail that is presented on the test description is slightly excessive.

-Results:

Comments: Add this subtitle: Phenotypic characteristics of the recovered isolates.

Response: thank you very much for your suggestions, we incorporate this subtitle 

Comments: Illustrate in detail the phenotypic characteristics of the recovered isolates, especially E. coli and K. pneumonia,

Response: Thank you. The general phenotypic characteristics of the recovered isolates espe-cially for E. coli and K. pneumoniae are described in the result sections on page 11, lines 263-280.

-Antimicrobial susceptibility testing:

Comments: Illustrate in a new table the occurrence of MDR (Multidrug Resistance) among the recovered isolates (illustrate the names of the antimicrobial classes and different antibiotics):

No. of strains%Type of resistance

R, MDR, and XDR Phenotypic multidrug resistance (Antimicrobial classes and different antibi-otics).

Response: Thank you for this suggestion. The antimicrobial classes of the tested antimicro-bial agents are indicated in Table 2 for each type of isolated, while the MDR, is presented in table 3 on page 14. But for testing XDR, and PDR, we agree with potential limitations be-cause we did not test all antimicrobial from the antimicrobial class of the drugs.

Comments: The antibiotic-resistance genes

Response: We agree that this is a potential limitation of the study. We have added this as a limitation and stated as: “Antibiotic resistance encoding genes (ARGs) of isolates were not detected as a confirmatory test due to a lack of molecular techniques and primers”.

Comments: Increase the resolution of all figures (it should be 600 dpi).

Response: thank you very much. We tried to increase the resolution using the PACE digital diagnostic tool.

-Discussion: 

Comments: The authors are advised to illustrate the real impact of their findings without repeti-tion of results.

Response: Thanks for your kind reminders. We revised the discussion as follows (page 25 and 26, lines 365-417)

Comments: Illustrate the different mechanisms of antimicrobial resistance in E. coli and K. pneumonia.

Response: Thanks for your kind reminders. We tried to add some point’s mechanisms of antimicrobial resistance in E. coli and K. pneumonia on pages 25 and 26, lines 387-392, page, 27 line 418-421).

-Conclusion

Comments: This should be rephrased to be sounded. A real conclusion should focus on the ques-tion or claim you articulated in your study, which resolution has been the main objective of your paper?

Response: We Revised according to the amended title of the manuscript

6. PLOS authors have the option to publish the peer review history of their article (what does this mean?). If published, this will include your full peer review and any attached files.

If you choose “no”, your identity will remain anonymous but your review may still be made pub-lic.

Do you want your identity to be public for this peer review? For information about this choice, including consent withdrawal, please see our Privacy Policy.

Reviewer #1: No

Reviewer #2: No

Response: Yes

While revising your submission, please upload your figure files to the Preflight Analysis and Conversion Engine (PACE) digital diagnostic tool, https://pacev2.apexcovantage.com/. PACE helps ensure that figures meet PLOS requirements. To use PACE, you must first register as a us-er. Registration is free. Then, log in and navigate to the UPLOAD tab, where you will find de-tailed instructions on how to use the tool. If you encounter any issues or have any questions when using PACE, please email PLOS at figures@plos.org. Please note that Supporting Infor-mation files do not need this step.

Response: Thank you for your reminder about the figure format and after confirmation, we checked and submitted them.

---

## [Decision Letter · Decision Letter 1]

18 Feb 2022

High prevalence of fecal carriage of Extended-spectrum beta-lactamase and carbapenemase-producing Enterobacteriaceae among food handlers at the University of Gondar, Northwest Ethiopia

PONE-D-21-38463R1

Dear Dr. Amare,

We’re pleased to inform you that your manuscript has been judged scientifically suitable for publication and will be formally accepted for publication once it meets all outstanding technical requirements.

Kind regards,

Abdelazeem Mohamed Algammal, Prof, Ph.D

Academic Editor

PLOS ONE

Additional Editor Comments (optional):

Reviewers' comments:

Reviewer's Responses to Questions

**Comments to the Author**

1. If the authors have adequately addressed your comments raised in a previous round of review and you feel that this manuscript is now acceptable for publication, you may indicate that here to bypass the “Comments to the Author” section, enter your conflict of interest statement in the “Confidential to Editor” section, and submit your "Accept" recommendation.

Reviewer #2: All comments have been addressed

2. Is the manuscript technically sound, and do the data support the conclusions?

Reviewer #2: Yes

3. Has the statistical analysis been performed appropriately and rigorously? 

Reviewer #2: Yes

4. Have the authors made all data underlying the findings in their manuscript fully available?

Reviewer #2: Yes

5. Is the manuscript presented in an intelligible fashion and written in standard English?

Reviewer #2: Yes

6. Review Comments to the Author

Reviewer #2: The authors have carried out significant changes to the manuscript. They have addressed all the suggested corrections and comments. Really, it's an interesting study that has a significant impact. Now, the manuscript could be accepted.

Congratulations.

7. PLOS authors have the option to publish the peer review history of their article (what does this mean?). If published, this will include your full peer review and any attached files.

Reviewer #2: No

---

## [Editor Report · Acceptance letter]

8 Mar 2022

PONE-D-21-38463R1 

High prevalence of fecal carriage of Extended-spectrum beta-lactamase and carbapenemase-producing Enterobacteriaceae among food handlers at the University of Gondar, Northwest Ethiopia. 

Dear Dr. Amare:

I'm pleased to inform you that your manuscript has been deemed suitable for publication in PLOS ONE. Congratulations! Your manuscript is now with our production department. 

Kind regards, 

on behalf of

Professor Abdelazeem Mohamed Algammal 

Academic Editor

PLOS ONE